# Effect of Herbal Feed Additives on Goat Milk Volatile Flavor Compounds

**DOI:** 10.3390/foods12152963

**Published:** 2023-08-05

**Authors:** Jacek Antoni Wójtowski, Małgorzata Majcher, Romualda Danków, Jan Pikul, Przemysław Mikołajczak, Marta Molińska-Glura, Joanna Foksowicz-Flaczyk, Agnieszka Gryszczyńska, Zdzisław Łowicki, Karolina Zajączek, Grażyna Czyżak-Runowska, Maria Markiewicz-Kęszycka, Daniel Stanisławski

**Affiliations:** 1Department of Animal Breeding and Product Quality Assessment, Faculty of Veterinary Medicine and Animal Science, Poznań University of Life Science, 60-637 Poznań, Poland; grazyna.czyzak-runowska@up.poznan.pl; 2Laboratory of Research on Volatile and Sensorally Active Compounds, Faculty of Food Science and Nutrition, Poznań University of Life Sciences, 60-624 Poznań, Poland; malgorzata.majcher@up.poznan.pl; 3Department of Dairy and Process Engineering, Faculty of Food Science and Nutrition, Poznań University of Life Sciences, 60-624 Poznań, Poland; romualda.dankow@up.poznan.pl (R.D.); jan.pikul@up.poznan.pl (J.P.); 4Department of Pharmacology, Poznań University of Medical Sciences, 60-806 Poznań, Poland; przemmik@ump.edu.pl; 5Department of Forest Economics and Technology, Faculty of Forestry and Wood Technology, Poznań University of Life Sciences, 60-637 Poznań, Poland; marta.glura@up.poznan.pl; 6Department of Innovative Biomaterials and Nanotechnologies, Institute of Natural Fibres and Medicinal Plants, 60-630 Poznań, Poland; joanna.flaczyk@iwnirz.pl; 7Department of Pharmacology and Phytochemistry, Institute of Natural Fibres and Medicinal Plants, 62-064 Plewiska, Poland; agnieszka.gryszczynska@iwnirz.pl (A.G.); zdzislaw.lowicki@iwnirz.pl (Z.Ł.); karolina.zajaczek@iwnirz.pl (K.Z.); 8School of Agriculture and Food Science, University College Dublin, Belfield, D04 V1W8 Dublin 4, Ireland; maria.markiewicz-keszycka1@ucd.ie; 9Computer Laboratory, Faculty of Veterinary Medicine and Animal Science, Poznań University of Life Science, 60-637 Poznań, Poland; daniel.stanislawski@up.poznan.pl

**Keywords:** goat milk, herbal feed additives, volatile flavor compounds, milk taste, milk smell

## Abstract

The aim of this study is to investigate the effects of herbal supplements administered to goats on sensory quality and volatile flavor compounds in their milk. The experiment was conducted on sixty Polish white improved goats randomly allocated into five feeding groups (four experimental and one control) of twelve goats each. The trial lasted 12 weeks. The experimental animals received supplements containing a mixture of seven or nine different species of herbs at 20 or 40 g/animal/day. The control group received feed without any herbal supplements. Milk obtained from experimental and control groups of animals was characterized by a low content of aroma compounds, with only 11 chemical compounds being identified. Decanoic methyl ester, methylo 2-heptanone and methylo-butanoic methyl ester had the highest share in the total variability of the tested aroma compounds (PCA). During the sensory evaluation, the smell and taste of most of the samples were similar (*p* > 0.05). However, the addition of herbal feed supplements lowered the concentration of Caproic acid (C6:0), Caprylic acid (C8:0) and Capric acid (C10:0), which caused a significant reduction in the goaty smell of milk. The obtained results indicate that the studied herbal supplements can reduce the intensity of goaty smell and allow goat milk production without modification of other sensory features.

## 1. Introduction

Humans have consumed goat milk for centuries, and, despite the passage of time, it still plays a vital role in the food culture of many regions. After buffalo milk, it is the most produced/consumed non-cow milk in the world. Its production has been on an upward trend in recent decades and is projected to increase by 53% by 2030 [1,2]. The growing demand for goat milk, especially in Western countries, is largely due to its health-promoting properties. Fat globules and casein micelles in goat milk are smaller than in cow milk; moreover, it has a higher concentration of short- and medium-chain fatty acids. Due to these features, it is more easily absorbed and has a lower allergenicity than cow milk [3,4,5].

Natural methods of increasing the concentration of biologically active components in milk play an increasingly important role for dairy goat producers, consumers and researchers. The supplementation of dairy goats with feeds, such as false flax cake or rolled rape seed, increase the share of polyunsaturated fatty acids, particularly CLA, and is an example of such a method [6]. The addition of phytobiotic feed additives to the animals’ diet to improve the health-promoting properties of milk represents another research trend. Phytobiotics are substances classified as sensory additives intended to improve the aroma and/or taste of feed [7]. Phytobiotic feed supplements are produced from wild plants or field crops [8]. Phytobiotics are produced from the parts of plants with a sufficiently high accumulation of active substances. These include leaves, rhizomes, roots, flowers, bark, fruits or seeds. Plant-stimulating or prophylactic and therapeutic properties depend on the concentration of biologically active substances, vegetation phase at the harvest, place of the harvest, weather conditions and method of drying and storage [9,10]. Post-production waste from the herbal industry can also be used as feed additives, provided that they still have the appropriate content of active substances (milk thistle endosperm) [11].

Studies conducted on goats showed that herbal supplements have a positive effect on the digestive processes of the goats and increasing the colonies of lactic acid bacteria (LAB) that fortify microbial homeostasis in the gastro-intestinal tract [12]. Other results revealed a significant effect of herbal supplements on the concentration of biologically active substances, such as apigenin, apigenin 7-glucoside and chlorogenic acid, in goat serum [13]. Another interesting result related to modifying goat milk’s fatty acid profile by adding fenugreek grain to goats’ diet [14]. Phytobiotics as pure botanicals, such as thymol and vanillin, when used in the nutrition of dairy goats, have confirmed a positive effect on milk yield and animal metabolic and immune status [15].

Milk produced by goats that have been administered with natural supplements has also been evaluated regarding its quality and sensory characteristics. Most studies, however, focused on the impact of ruminants’ diet on the volatile flavor compounds, mainly in ripening cheeses made from cow milk [16]. Stefanon et al. [17] found the effect of grass silage on the increase in keto odor compounds (3-butanedione, 2-butanone, 2-hexanone, 2-heptanone and 2-methyl-1-butanol) in Montasio ripening cheeses. A few studies on goats confirm that feed administrated to animals can impact the aromatic profile of milk. Bennato et al. [18] found a significant impact of dried licorice root on the decrease in the concentration of hexanoic acid. Cais-Sokolińska et al. [19] examined the dairy products obtained from the milk of goats receiving the feed additive DDGS and found less prevalent whey and more prominent cream aroma in kefirs produced from the experimental milk.

This study aims to investigate the effect of herbal feed additives on the sensory quality and volatile compounds of goat milk. We hypothesized that the herbal feed additives in the goats’ diet may have an impact on milk aroma quality. To the best of our knowledge, this is the first detailed identification of odorants in goat milk affected by herbal feed additives.

## 2. Materials and Methods

### 2.1. Ethical Approval

All procedures used in this study were approved by the Local Bioethics Committee (Poznań, Poland; decision no. 57/2020).

### 2.2. Geographical Orientation and Animals

Sixty clinically healthy, Polish white improved goats were selected from a herd at a specialized farm located in the Wilekopolska region at Bukowiec, Poland (52°51′41″ N; 16°52′12″ E). The somatic cell count (SCC) measured immediately before the experiment (during the third week of lactation) was in line with expectations and did not exceed 800 × 10^3^/mL. Animals’ age and live weight ranged from 20–30 months and 56–60 kg, respectively. The goats were in their second lactation and entered the study at 28.1 ± 2.7 d in milk (DIM).

The animals were randomly assigned to five feeding groups of twelve goats each:Group I-20 (receiving 20 g of herbal supplement—a mix of seven herbs);Group I-40 (receiving 40 g of herbal supplement—a mix of seven herbs);Group II-20 (receiving 20 g of herbal supplement—a mix of nine herbs);Group II-40 (receiving 40 g of herbal supplement—a mix of nine herbs);Group CG (control group, no herbal supplements).

Goats received 20 g or 40 g of herbal supplement per day. To enable the identification, goats were wearing electronic transponders and collars with numbers.

Milk was obtained in the milking parlor at 0530 and 1730 with pre-milking and post-milking disinfection of udders and teats [20].

The milking of each goat was preceded by for-stripping and udder and teats cleaning. Wet udder paper towels (Biocell™, Delaval, Tumba, Sweden) were used for cleaning prior to milking, and 20% solution of iodine agent Dipal™ Conc (Delaval, Tumba, Sweden) was used for post-milking teats’ dipping.

The batch milk samples from daily milking were collected at the experiment beginning (wk 0) and at the end of 1, 4, 8 and 12 weeks of the experiment.

### 2.3. Herbal Supplements

The herbal supplements used in the experiment were composed of 7 or 9 herbs—in the paper referred to as herbal supplement I and herbal supplement II, respectively.

The herbal mix I contained common nettle, dried flowering shoot tips of common agrimony, fruit of caraway, fruit of coriander, seeds of fenugreek, plantain and bark of purple willow.

Herbal mix II comprised different amounts of herbs that were also used in herbal mix I, i.e., common nettle, dried flowering shoot tips of common agrimony and fruit of coriander, plus 6 other herbs: fennel (fruit), peppermint (leaves), chamomile (flower clusters), milk thistle (endosperm) and thyme (leaves).

A detailed composition of herbal components can be found in our patent applications (Polish Patent Office submissions P.4334426 and P.433779).

### 2.4. Animal Nutrition

Goats’ diets were formulated to meet their nutritional needs [21]. Animals received their feed once a day in the form of the total mix ration (TMR). Experimental concentrate with the herbal mix was excluded from TMR and offered to animals separately. Goats’ had ad libitum access to fresh water and mineral salt lick. Detailed information on the composition of the applied rations is presented in the paper of Foksowicz-Flaczyk et al. (2022)s [12].

The pelleted experimental concentrate was composed of cereal grains, rapeseed meal and sunflower meal and herbal supplements. The control group (CG) received concentrate without herbal supplements. Depending on the experimental group, goats received 300 g of concentrate plus:A total of 20 g DM of Herbal Mix 1 (6.6 g 100 g^−1^ concentrate dry matter)—Group I-20;A total of 40 g DM of Herbal Mix 1 (13.2 g 100 g^−1^ concentrate dry matter)—Group I-40;A total of 20 g DM of Herbal Mix 2 (6.6 g 100 g^−1^ concentrate dry matter)—Group II-20;A total of 40 g DM of Herbal Mix 2 (13.2 g 100 g^−1^ concentrate dry matter)—Group II-40.

The detailed composition of the experimental concentrates is presented in the work of Foksowicz-Flaczyk et al. [12].

### 2.5. Determination of Fatty Acid Composition

The fatty acid composition of the bulk milk samples collected in the 12th week of the experiment was determined using capillary gas chromatography. Milk fat was extracted with chloroform:methanol (2:1, vol/vol) using the method described by Folch et al. [22]. The fatty acid composition in the milk was estimated using methyl esters prepared by direct transesterification, according to the IUPAC method [23]. The analyses were performed using a Hewlett Packard model 6890NGC (Agilent Technologies, Palo Alto, CA, USA) equipped with a flame ionization detector, autosampler and split/splitless injector. Separations were performed on a BPX70 column (60 m × 0.22 mm i.d. 0.25 × µm film thickness, stationary phase 70% cyanopropylpolysilphenylene-siloxane; SGE Analytical Science, Austin, TX, USA). The conditions for the chromatographic analysis were as follows: The injector temperature was 230 °C with a split ratio set to 100:1 and the FID temperature 270 °C. The oven temperature was ramped from 130 °C (3 min) to 235 °C (5 min) at a rate of 2 °C min^−1^. Helium was used as a carrier gas with a constant pressure of 40 psi at a flow rate of 0.3 mL min^−1^ and an injection volume of 1 µL. The fatty acid content is expressed as a percentage of the identified fatty acids.

### 2.6. Volatile Compounds

#### 2.6.1. Isolation Method

Solid-phase microextraction (SPME) was used to isolate volatiles. For each analysis, 10 mL of milk sample was placed in 20 mL headspace vials, spiked with 1 µg of the internal standard naphthalene d8, and capped with PTFE/silicon septa caps. Volatiles were extracted with 2 cm CAR/PDMS/DVB fibre (Supelco, St. Louis, MO, USA) conditioned before analyses according to the manufacturer’s recommendations. To avoid the formation of artifacts during milk heating, extraction was performed from the tray at 22 °C for 40 min. All analyses were executed using an MPS autosampler (Gerstel, Linthicum, MD, USA).

#### 2.6.2. Gas Chromatography/Mass Spectrometry

The volatile chemical compounds were identified using Agilent Technologies 7890A GC coupled to a 5975C MSD with an SLB-5MS (25 m × 0.2 mm × 0.33 µm) column. Operating conditions for GC/MS were as follows: helium flow, 32.2 cm/s; oven conditions: initial oven temperature 40 °C (1 min), raised at 9 °C/min to 180 °C and at 20 °C/min to 280 °C. Mass spectra were recorded in an electron impact mode (EI, 70 eV) in a scan range of 33–350 *m*/*z*. All the volatiles’ identification was performed by the comparison of their mass spectra and retention indices (RI) to those of standard compounds, National Institute of Standards and Technology (NIST) 09 library of mass spectra, and the literature data. Retention indices were calculated for each compound using homologous C7–C24 n-alkanes series.

#### 2.6.3. Data Quantitation

The relative concentrations of the investigated compounds were calculated by relating the peak areas of the internal standard, ^2^H_8_ naphthalene, to the peak areas of the compounds of interest. The presented results represent semi-quantitative data and were used for comparison to show the difference in the concentration levels of volatiles in different milk samples.

### 2.7. Sensory Evaluation of Milk Samples

The sensory evaluation of milk samples was performed using the profiling method of quantitative descriptive analysis [24].

An assessment of smell and taste intensity (from 0—‘imperceptible’ to 5—‘very intense’) was performed. The following discriminants of taste were taken into account: milky, sour, herbal, bitter, salty, and pleasant. The discriminants of smell were: milky, sour, herbal, goaty and pleasant. These discriminants were established by an eight-person review panel during a special session. Sensory quality assessment was performed on samples on the day following their collection in the 12th week of the experiment (T12). Milk from all animal groups was evaluated in three series—sessions (*n* = 120; eight panelists × five groups of milk × three series of evaluation).

The sensory panel consisted of 8 people (4 women and 4 men, mean age 53.4 ± 10.9) with several years of experience in milk quality assessment. The evaluators were characterized by high sensory sensitivity and were trained in the sensory evaluation of goat milk and products in accordance with the standard ISO 8586 [25]. The training schedule of the sensory panel included identification and description of the milk flavor and procedures that used the response scale.

The test room conditions, sample preparation, sensory evaluation and results discussion met the requirements of the ISO 8589 standard [26]. The sensory evaluation was performed in an air-conditioned sensory laboratory where the temperature was 22 °C. The evaluators had free access to drinking water. Subsequent samples were evaluated after rinsing the mouth when the taste residues from the previous sample had been removed from the palate.

### 2.8. Statistical Snalysis

The intensity of goat milk taste and smell were analyzed with SAS 9.4 2019 (SAS Institute Inc., Cary, NC, USA). The MEANS and UNIVARIATE procedures were used to calculate the necessary elements of descriptive statistics (mean, standard deviation).

The GLM procedure was used to estimate the influence of the analyzed experimental factors, the type of mixture and the dose on the characteristics of sensory evaluation, according to the following linear model:Y_ij_ = µ+ M_i_ + e_ij_

Y_ij_: phenotypic value of the analyzed trait;

µ: population average;

M_i_: constant effect of the type of mixture–dose (i = 1, 2, 3, 4 and 5);

e_ij_: random error.

The results of the volatiles’ analysis were subjected to PCA (principal component analysis). The analysis was conducted in the Statistica software package [27] at the significance level α = 0.05.

## 3. Results and Discussion

### 3.1. Basic Chemical Composition and Hygienic–Cytological Quality of Bulk Tank Milk Samples at the 12th Week of the Experiment

The chemical composition and hygienic quality of the bulk milk samples in the 12th week of the experiment are presented in Table 1. The milks obtained from animals from Groups I-40, II-40 and I-20 were characterized with the highest level of fat—4.36, 4.35 and 4.30%, respectively. The milk of the control group (CG) had the lowest fat concentration and the highest lactose content, at 4.46%. The protein content was similar for all groups, 3.34–3.43% (except Group II).

The results of the basic milk composition obtained in the study are consistent with the values for this stage of lactation recorded by other authors [5,28].

The milk of all groups of animals was characterized by good hygienic and cytological quality. Total bacteria count (TBC) did not exceed 43.26 × 10^3^/cm^3^ (log_10_TBC = 4.61) and was within the range of values obtained in [29,30], where log_10_TBC = 4.06–4.98.

The cytological quality of the examined milk was also very good. The somatic cell count in the milk of all groups was relatively low, from 397.78 to 426.11 × 10^3^/cm^3^. According to Paape et al. [31], the milk of clinically healthy goats is characterized by SCC within the range of 270 × 10^3^–2000 × 10^3^ cells/mL. In Silanikove et al. [32], the milk of clinically healthy goats was characterized by a cytological quality of 350 × 10^3^ cell/mL. Additionally, Leitner et al. [33] found a similar level of somatic cells in goat milk samples, 417 × 10^3^ cell/mL, as presented in this study.

### 3.2. Fatty Acid Profile of Bulk Tank Milk Samples

The fatty acid profile of the bulk tank milk samples is presented in Table 2.

The milk of CG was characterized by a higher proportion of saturated fatty acids (SFA), 71 g/100 g, of total fatty acids compared to the milk of animals receiving herbal supplements. The milk obtained from the experimental groups I-20 i I-40 had 69 g/100 g and 66.6 g/100 g of SFA, respectively. A similar level of SFA was determined in the milk obtained from groups II-20 and II-40, at 69.0 and 66.6 g/100 g, respectively. The experimental groups I and II had a lower concentration of medium-chain saturated fatty acids C6:0, C8:0 and C10:0 compared to CG.

The experimental milk contained more unsaturated fatty acids (UFAs) than CG. The range of values for monounsaturated acids (MUFAs) was 26.1–29.0 g/100 g and for polyunsaturated acids (PUFAs) 4.3–4.5 g/100 g. In the CG, the concentration of MUFAs was 25.1 g/100 g and the PUFA 3.9 g/100 g.

The fatty acid (FA) profile of the milk samples analyzed in this study was consistent with the FA profile presented by other authors. In their literature review, Kęszycka et al. [34] reported the average level of fatty acids for goat milk as follows: SFAs—68.79 g/100 g, MUFAs—24.48 g/100 g and PUFAs—3.70 g/100 g; however, the FA profile of goat milk can be affected by their diet. For example, pasture forage and the supplementation of dairy goats with various feed additives (false flax, extruded linseed and artichoke plant) significantly increase the proportion of polyunsaturated fatty acids, in particular, CLAs [6,19,34,35,36].

The specific taste and aroma of goat milk is caused by three medium-chain fatty acids: caproic (C6:0), caprylic (C8:0) and capric (C10:0) acids [37,38]. Their share in the fatty acid profile of goat milk can be up to about 15%, while cow’s milk usually contains no more than 5% [39]. The concentration of these fatty acids in goat milk also depends on their breed. Šlyžius et al. [40] presented results that indicate that the concentration of C8:0 and C10:0 was significantly higher in Alpine goats’ milk than in milk obtained from Saanen goats [40]. Moreover, Vulić et al. [41], who also examined the milk of Alpine and Saanen goats, also reported the effect of breed on the concentration of caproic acid (C6:0).

### 3.3. Volatile Compounds in Goat Milk

The concentration of aroma compounds in goat milk during the 12-week administration of herbal mixtures is summarized in Table 3. Overall, the examined milk was characterized by a low content of aromatic compounds. Eleven chemical compounds were identified, five of which belonged to the group of acids: acetic acid (AA), pentanoic acid (PP), hexanoic acid (HA), octanoic acid (OA) and decanoic acid (DA). Two compounds represented the group of methyl ketones: 2-heptanone (2H) and 2-nonanone (2N). Four compounds represented the methyl esters group: butanoic methyl ester (BM), hexanoic methyl ester (HM), octanoic methyl ester (OM) and decanoic methyl ester (DM).

AA and PA were not identified in the milk obtained from CG animals. In the milk of the experimental animals, AA was identified only in the 12th week of the experiment. AA concentration in milk samples I-20 and I-40 was several times lower compared to the milk of goats receiving herbal mixture II. PA in the milk of animals from groups I-20 and I-40 was identified from the 4th week, and in the milk of II-20 and II-40 animals, only in the last, 12th week of the experiment. Its concentration in II-40 milk samples was significantly higher than that recorded for I-40 milk. Acetic acid is a short-chain fatty acid that is naturally present in goat milk. It is formed from the breakdown of carbohydrates by lactic acid bacteria in the milk, which are a type of beneficial bacteria that are found in many fermented foods, such as yogurt, kefir and sauerkraut. Acetic acid has a sour taste and a pungent smell. It is responsible for the characteristic sour taste of fermented goat milk products, such as yogurt and kefir. Acetic acid is also a natural preservative, which helps to keep fermented goat milk products fresh [42,43]. Pentanoic acid with cheesy and milk aroma has been recently described as the key odorant in goat milk with an odor activity value of 3 [44].

Hexanoic acid (HA), octanoic acid (OA) and decanoic acid (DA) were present in all milk samples, regardless of the period of their collection (Figure 1, Figure 2 and Figure 3). The lowest concentration of these acids was identified in the milk of animals from the CG. The milk from the I-40 group showed lower concentrations of HA, OA and DA compared to I-20 milk. These fatty acids are formed from the breakdown of milk fat by the lipolytic bacteria present in the milk. The amount of hexanoic and octanoic acids in goat milk can vary depending on the type of lipolytic bacteria that are present, the length of time that the milk is stored for and the temperature at which the milk is stored. In general, the longer the milk is stored, the higher the concentration of hexanoic and octanoic acids will be. Hexanoic and octanoic acids have a strong, goaty odor. They are responsible for the characteristic goaty flavor of goat milk. Hexanoic and octanoic acids are also a natural preservative, which helps to keep goat milk fresh [43,45].

This relationship, although not so visible, was also observed when examining the milk samples of animals of nutritional group II.

Within the methyl ketones, an increase in 2-heptanone (2H) and 2-nonanone (2N) was observed over time. The highest level of 2N was observed in I-20 and I-40 milk sampled in the 12th week of the experiment. In general, methyl ketones in dairy products can be formed during the enzymatic lipolysis of the free fatty acids that are oxidized into 3-oxo acids. Additionally, their formation can be accelerated by high temperatures and the presence of oxygen. They are known to be responsible for the aroma formation of mold-ripened cheeses, such as Gorgonzola cheese and several French blue cheeses [46,47]. Their odor descriptors are characterized as fruity, floral and musty with such ketones as 2-octanone, 2-nonanone, 2-decanone and 2-undecanone, whereas blue cheese notes are attributed to 2-heptanone [48].

The content of methyl esters increased during the experiment for all nutritional groups (Figure 4, Figure 5, Figure 6 and Figure 7). The herbal supplement increased the concentration of this group of compounds in milk. The concentration of methyl esters in milk was the highest in the milk obtained from animals receiving 40 g of the supplement daily (I-40 and II-40). According to [44], esters represent the largest group of volatiles present in goat milk. They also reported that ethyl hexanoate and methyl hexanoate belong to the key odorants of goat milk with an odor activity value calculated as 35 and 20, respectively. In our studies, only methyl esters were identified, which are responsible for the fruity and sweet odor attributes. They are formed in the course of esterification, which is catalyzed by enzymes called lipases produced by bacteria in milk. It is a natural process that occurs during the storage or processing of milk. Esters are responsible for the characteristic flavor of many dairy products, such as butter, cheese and yogurt [44,49].

The principal component analysis (PCA) is a technique mainly employed in data dimensionality reduction and pattern recognition [50]. The aim of PCA is to simplify the analysis of data consisting of numerous variables, while retaining the variation occurring in the dataset. The grouped experimental combinations resulting from PCA are shown in Figure 8.

The principal components were determined as a linear combination of the observed variables. The calculations determining the principal components allowed us to determine the eigenvalues and the corresponding eigenvectors. The eigenvalue explains how much of the total variability is represented by a given principal component. The first principal component explains the largest part of the variance, the second principal component explains the largest part of the variance not explained by the previous component, and the next principal component accounts for the largest part of the variance not explained by the previous two components [50,51]. As a result, each subsequent principal component explains a successively smaller part of the variance, i.e., successive eigenvalues are smaller and smaller. The calculated eigenvalues of the components are presented in Table 4. In addition to the percentage value of individual components, Table 4 also includes their cumulative value and cumulative % variability. The combined value of the first two principal components was relatively high, accounting for 79.75% of the total variability (Table 4). Thus, they represent all aroma compounds considered in the paper [50,51].

The factorial coordinates of the variables, estimated based on correlations, are presented in Table 5. The values with the highest share in the total variability of the tested aroma compounds are marked in bold, and they include: decanoic methyl ester, methyl 2-heptanone and methyl-butanoic methyl ester (Table 5). Among chemical compound groups, methyl-esters had the largest share in the total variability, and acids had the smallest share (Table 5).

The results from Table 4 and Table 5 are also presented in the chart in Figure 9. It is a graphical representation of the interrelationships between the components, primary variables and cases, providing a holistic view of the principal components analysis. Figure 9 shows the vectors connected to the origin of the coordinate that represents the original variables. These vectors are placed on a plane defined by two selected principal components: Factor 1 and Factor 2, with a total variability of 50% and 29.75%, respectively. The length of the vector contains information about the primary variable carried by the principal components determining the coordinate system. The longer the vector, the greater the contribution of the original variable to the structure of the components. Loads are correlations between original variables and components, and their maximum value does not exceed 1. In the current study, the longest vectors were noted for methyl esters: hexanoic methyl ester, octanoic methyl ester, decanoic methyl ester and hexanoic acid (Figure 9).

The negative values of the coordinates of the end of the vector, indicating a negative correlation between the primary variable and the principal components forming the coordinate system, were observed for all four identified methyl esters and kethon-2-nonanone (Figure 9).

An important information in the chart is the degree of correlation of the primary variables (Figure 9). It is established based on the angle of inclination of the original variable vectors. The inclination angle between vectors within a range of 0° < α < 90° indicates a positive correlation between these variables. The smaller the angle of inclination, the stronger the correlation. In the case of the studied variables, particularly high, positive correlation values were noted between the hexanoic, octanoic, decanoic and acetic acids. A similarly high positive correlation was found between octanoic methyl ester and decanoic methyl ester (Figure 9). However, no correlation was found between methyl esters (hexanoic methyl ester, methyl-butanoic methyl ester and octanoic methyl ester) and acids (hexanoic, octanoic, decanoic and acetic acids), which is indicated by a right angle (α = 90°) between the vectors represented by these groups.

### 3.4. Sensory Quality of Milk

The sensory assessment of the goat milk taste and smell was conducted on milk samples taken in the last, 12th week of the experiment (Table 6). The milk obtained from all feeding groups was characterized by a very faint smell. The smell and taste of the majority of milk samples were similar (*p* > 0.05), except for the intensity of the goaty and pleasant smells, which varied between the feeding groups. The most intense goaty smell was identified in the milk obtained from the CG (*p* < 0.0001). The addition of herbal feed supplements, regardless of their type (herbal mixture I or II) and the amount consumed (20 or 40 g/day), significantly reduced the goaty smell of milk. These results are consistent with the medium-chain FA profile C6:0, C8:0 and C10:0 in the experimental milk samples (Table 2). In the sensory analysis, the pleasant smell rating was obtained by the milk samples with a less intense goaty smell (*p* < 0.02). The results of the sensory evaluation are presented in the graph in Figure 10. From a practical point of view, the results of the assessment are very promising, as they allow the use of the pro-health effect of the herbal mixtures tested in this study [12,13] without significance, except for the reduced intensity of the goaty smell, modification of the taste and smell of the produced milk.

To date, considerable research has been published in the field of the sensory quality of dairy products produced with the addition of herbal components [52,53,54]. However, there are no studies evaluating the sensory attributes of milk produced with the use of herbal supplements in the diet of farm animals.

## 4. Conclusions

To the best of our knowledge, a detailed investigation of odorants in goat milk affected by herbal feed additives has not been previously investigated. The addition of a herbal feed supplements, regardless of their type (herbal mixture I or II) and amount consumed (20 or 40 g/day), caused a significant reduction in the intensity of the goaty smell, which was a consequence of the lower concentration of fatty acids C6:0, C8:0 and C10:0 compared to the milk of the control group. The volatile compound analysis indicated the presence of only 11 chemical compounds: acids (5 compounds), methyl ketones (2 compounds) and methyl esters (4 compounds). Decanoic methyl ester, methyl 2-heptanone and methyl-butanoic methyl ester had the highest share in the total variability of the tested volatile compounds (principal component analysis). The increase in acids, methyl ketones and esters in the experimental milk samples obtained from the supplemented animals did not significantly affect its sensory qualities compared to the control group. This may be due to the high perception threshold of these compounds. The results of this study confirm that tested herbal additives added to goats’ diet allow the production of milk with a decreased goaty smell, without the further modification of smell and taste of goat milk.

## Figures and Tables

**Figure 1 foods-12-02963-f001:**
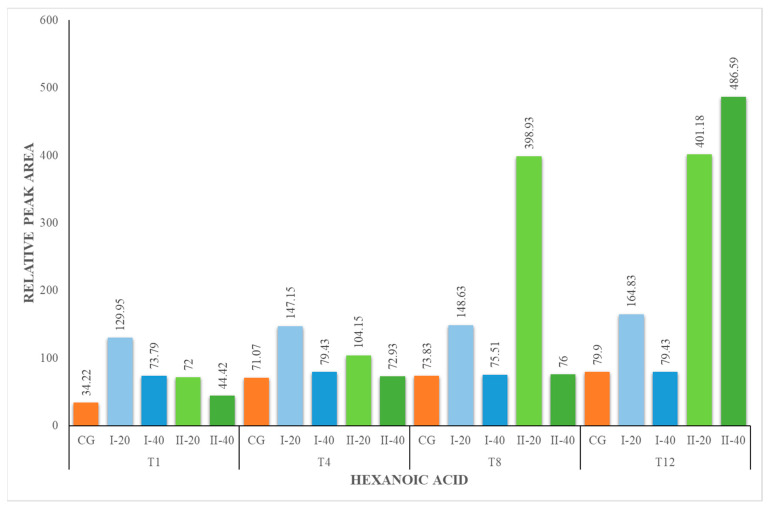
Concentration (relative peak area) of hexanoic acid at the 1st, 4th, 8th and 12th week of the experiment.

**Figure 2 foods-12-02963-f002:**
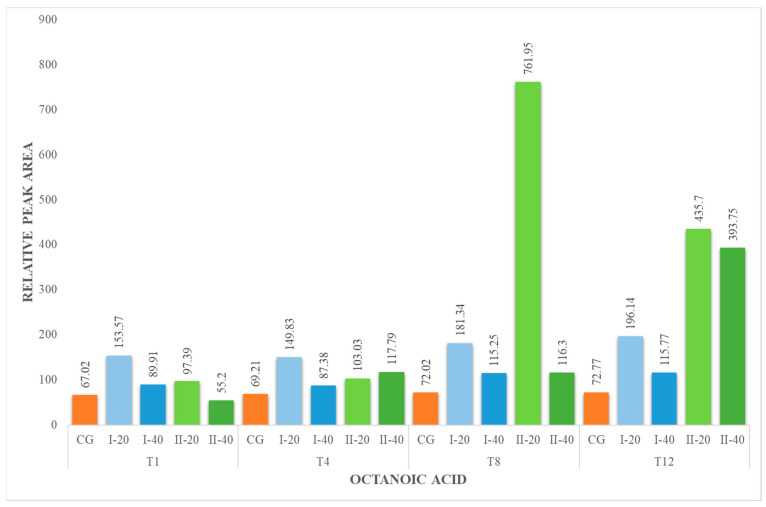
Concentration (relative peak area) of octanoic acid at the 1st, 4th, 8th and 12th week of the experiment.

**Figure 3 foods-12-02963-f003:**
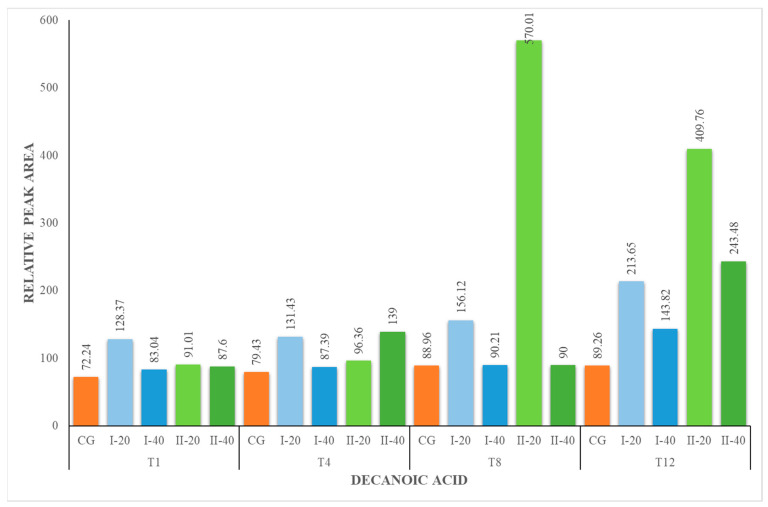
Concentration (relative peak area) of decanoic acid at the 1st, 4th, 8th and 12th week of the experiment.

**Figure 4 foods-12-02963-f004:**
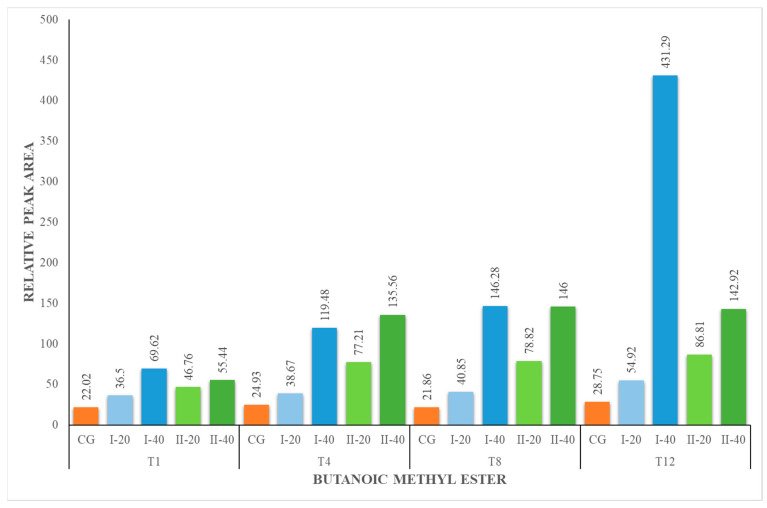
Concentration (relative peak area) of butanoic methyl ester at the 1st, 4th, 8th and 12th week of the experiment.

**Figure 5 foods-12-02963-f005:**
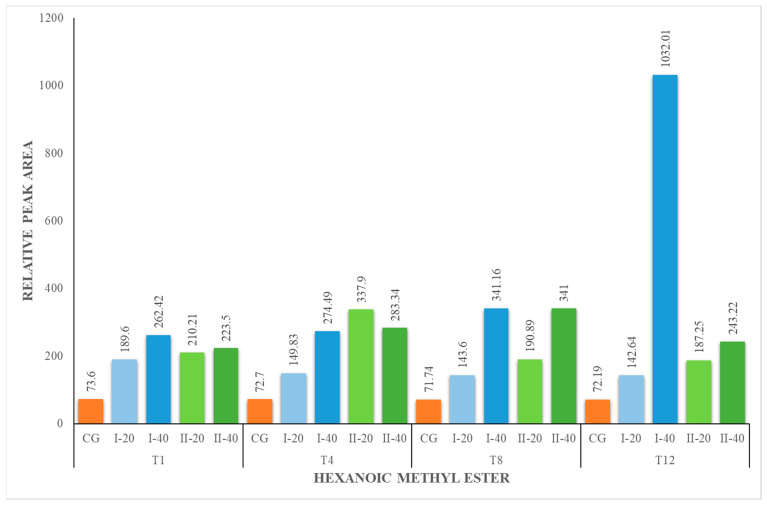
Concentration (relative peak area) of hexanoic methyl ester at the 1st, 4th, 8th and 12th week of the experiment.

**Figure 6 foods-12-02963-f006:**
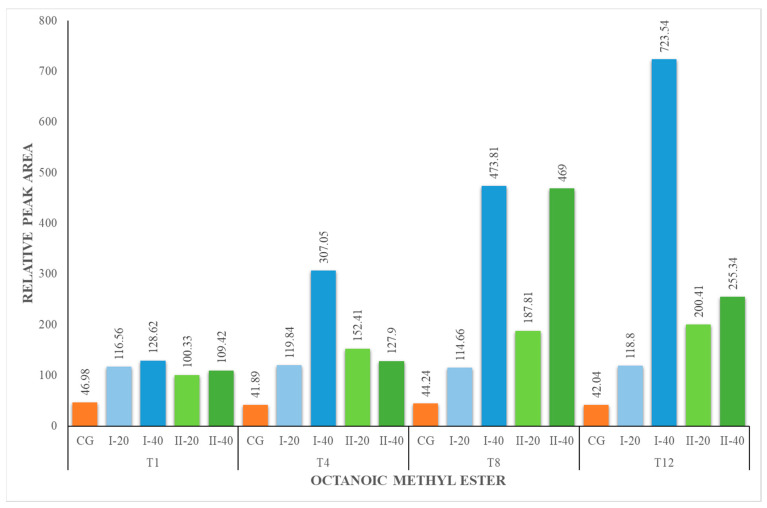
Concentration (relative peak area) of octanoic methyl ester at the 1st, 4th, 8th and 12th week of the experiment.

**Figure 7 foods-12-02963-f007:**
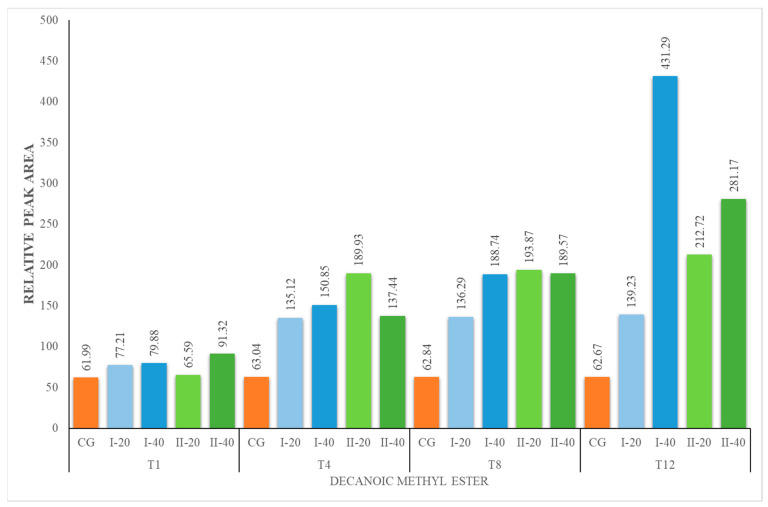
Concentration of decanoic methyl ester at the 1st, 4th, 8th and 12th week of the experiment.

**Figure 8 foods-12-02963-f008:**
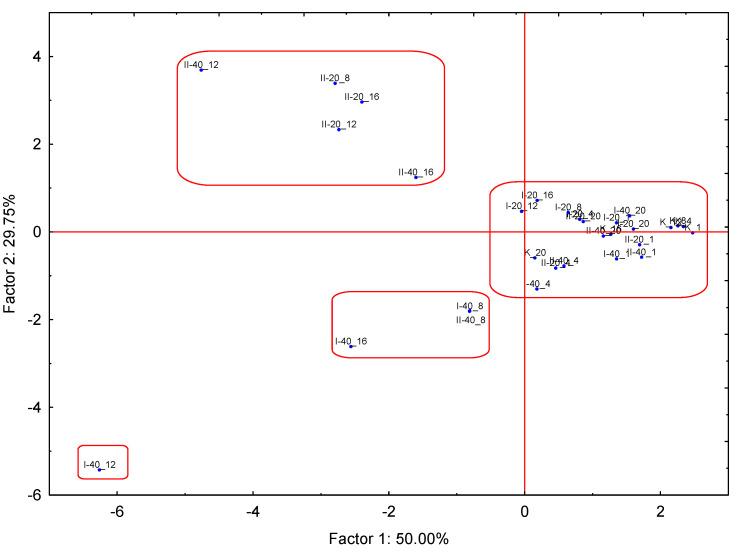
Grouped experimental combinations resulting from the application of PCA–PCA score plot of Factor 1 vs. Factor 2.

**Figure 9 foods-12-02963-f009:**
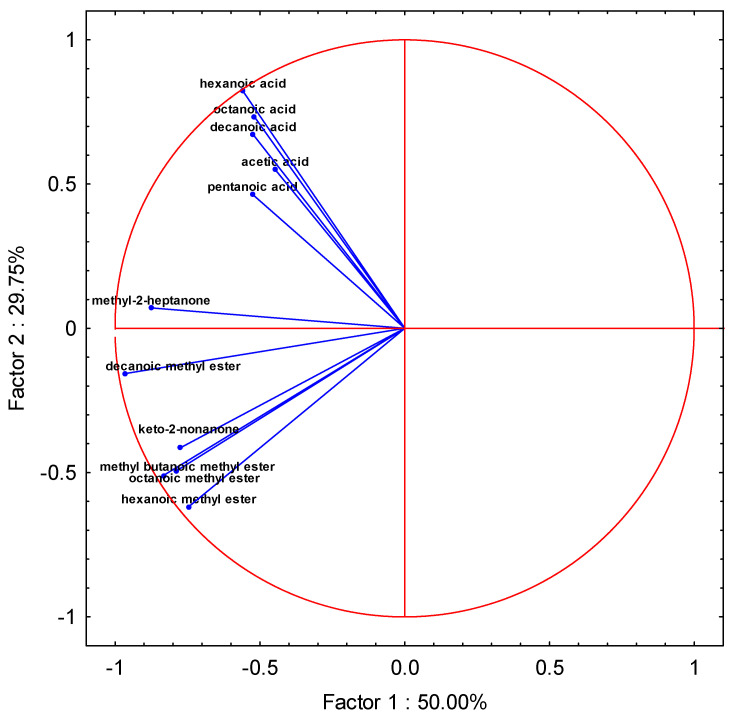
PCA of the volatile compounds determined in the bulk tank milk samples at the 12th week of the experiment.

**Figure 10 foods-12-02963-f010:**
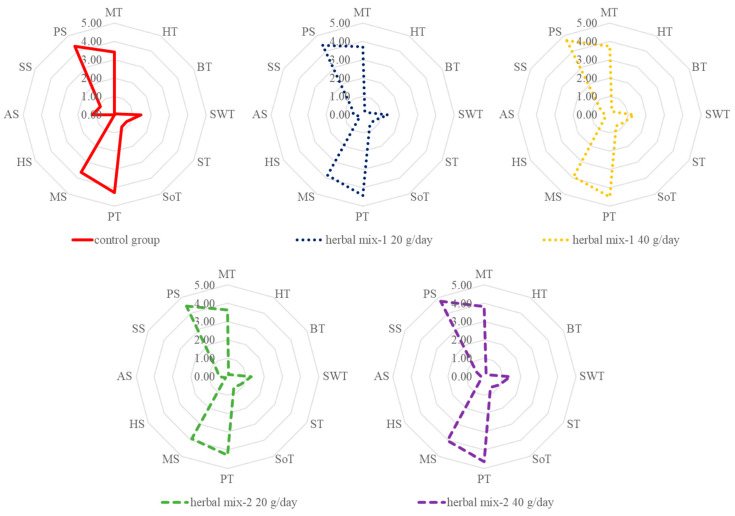
The sensory profile description of the evaluated milk samples. MT—milky taste; HT—herbal taste; BT—bitter taste; SWT—sweet taste; ST—salty taste; SoT—sour taste; PT—pleasant taste; MS—milky smell; HS—herbal smell; GS—goaty smell; SS—sour smell; PS—pleasant smell.

**Table 1 foods-12-02963-t001:** Basic chemical composition and hygienic–cytological quality of bulk tank milk samples at the 12th week of the experiment (mean ± SD).

Features	CG		I-20		I-40		II-20		II-40	
	$\bar{x}$	SD	$\bar{x}$	SD	$\bar{x}$	SD	$\bar{x}$	SD	$\bar{x}$	SD
Fat (%)	4.01	0.04	4.30	0.08	4.36	0.08	4.13	0.05	4.35	0.08
Protein (%)	3.41	0.04	3.34	0.03	3.51	0.04	3.43	0.04	3.58	0.05
Lactose (%)	4.46	0.05	4.32	0.05	4.41	0.04	4.38	0.05	4.38	0.05
TBC (10^3^/cm^3^)	41.18	2.35	43.26	3.05	39.22	1.61	40.62	2.05	41.78	2.16
SCC (10^3^/cm^3^)	397.78	11.31	426.11	13.62	387.38	11.75	406.44	12.67	411.34	12.34

Abbreviations: CG—control group; I-20—herbal mix I 20 g/day/animal; I-40—herbal mix I 40 g/day/animal; II-20—herbal mix II 20 g/day/animal; II-40—herbal mix II 40 g/day/animal; TBC—total bacteria count; SCC—somatic cell count.

**Table 2 foods-12-02963-t002:** Fatty acid (FA) composition of the goat bulk tank milk samples at the 12th week of the experiment (±SD).

FA, g/100 g of Total FA	CG	I-20	I-40	II-20	II-40
C4:0	1.7 ± 0.1	1.7 ± 0.1	1.6 ± 0.1	1.7 ± 0.1	1.6 ± 0.1
C6:0	2.2 ± 0.1	2.0 ± 0.1	2.0 ± 0.1	2.1 ± 0.1	1.9 ± 0.1
C8:0	2.6 ± 0.1	2.4 ± 0.1	2.3 ± 0.1	2.4 ± 0.1	2.3 ± 0.1
C10:0	9.7 ± 0.5	9.4 ± 0.5	8.5 ± 0.5	9.3 ± 0.5	8.6 ± 0.5
C10:1	0.2 ± 0.1	0.2 ± 0.1	0.2 ± 0.1	0.2 ± 0.1	0.2 ± 0.1
C12:0	4.7 ± 0.2	4.3 ± 0.2	3.8 ± 0.2	4.2 ± 0.2	3.7 ± 0.2
C12:1	0.1 ± 0.05	0.1 ± 0.05	0.1 ± 0.05	0.1 ± 0.05	0.1 ± 0.05
C13:0	0.1 ± 0.05	0.1 ± 0.05	0.1 ± 0.05	0.1 ± 0.05	0.1 ± 0.05
C14:0	11.0 ± 0.5	10.9 ± 0.5	9.8 ± 0.5	11.0 ± 0.5	9.9 ± 0.5
C14:1	0.3 ± 0.1	0.3 ± 0.1	0.3 ± 0.1	0.3 ± 0.1	0.3 ± 0.1
C15:0	1.1 ± 0.1	1.1 ± 0.1	1.0 ± 0.1	1.1 ± 0.1	1.0 ± 0.1
C15:1	0.2 ± 0.1	0.2 ± 0.1	0.2 ± 0.1	0.2 ± 0.1	0.2 ± 0.1
C16:0	28.6 ± 1.4	28.6 ± 1.4	27.2 ± 1.4	28.2 ± 1.4	27.4 ± 1.4
C16:1	1.2 ± 0.1	1.2 ± 0.1	1.3 ± 0.1	1.2 ± 0.1	1.3 ± 0.1
C17:0	1.4 ± 0.1	1.3 ± 0.1	1.7 ± 0.1	1.3 ± 0.1	1.6 ± 0.1
C17:1	0.3 ± 0.1	0.3 ± 0.1	0.3 ± 0.1	0.3 ± 0.1	0.3 ± 0.1
C18:0	7.7 ± 0.4	7.7 ± 0.4	8.5 ± 0.4	7.5 ± 0.4	8.4 ± 0.4
C18:1 *trans*	1.8 ± 0.1	2.0 ± 0.1	2.0 ± 0.1	2.0 ± 0.1	2.0 ± 0.1
C18:1 *cis-9*	19.4 ± 1.0	20.2 ± 1.0	22.8 ± 1.0	20.8 ± 1.0	22.9 ± 1.0
C18:1 *cis-11*	0.7 ± 0.1	0.7 ± 0.1	0.7 ± 0.1	0.7 ± 0.1	0.7 ± 0.1
C18:1n7	1.0 ± 0.1	1.0 ± 0.1	1.0 ± 0.1	1.0 ± 0.1	1.0 ± 0.1
C18:2	2.6 ± 0.1	2.8 ± 0.1	3.0 ± 0.1	2.8 ± 0.1	2.9 ± 0.1
C18:3 *n*-3 (ALA)	0.2 ± 0.1	0.2 ± 0.1	0.2 ± 0.1	0.2 ± 0.1	0.2 ± 0.1
C18:2 *cis9 trans11 (CLA*)	0.6 ± 0.1	0.7 ± 0.1	0.8 ± 0.1	0.7 ± 0.1	0.8 ± 0.1
C20:0	0.2 ± 0.05	0.1 ± 0.05	0.1 ± 0.05	0.1 ± 0.05	0.1 ± 0.05
C20:1	0.1 ± 0.05	0.1 ± 0.05	0.1 ± 0.05	0.1 ± 0.05	0.1 ± 0.05
C20:4 *n*-6	0.2 ± 0.1	0.3 ± 0.1	0.2 ± 0.1	0.3 ± 0.1	0.2 ± 0.1
C22:5 *n*-3	0.1 ± 0.05	0.1 ± 0.05	0.1 ± 0.05	0.1 ± 0.05	0.1 ± 0.05

**Table 3 foods-12-02963-t003:** Volatile compounds (VOC ± SD) of goat milk at the experiment’s beginning (T0), first (T1), fourth (T4), eighth (T8) and twelve (T12) weeks, obtained from goats fed a standard diet (CG) and with the dietary supplementation of herbal mixtures (I-20, I-40, II-20 and II-40).

VOC	T0	T1			T4			T8			T12		
	All Groups	CG	I-20	I-40	CG	I-20	I-40	CG	I-20	I-40	CG	I-20	I-40
ACI													
AA	0.00 ± 0.00	0.00 ± 0.00	0.00 ± 0.00	0.00 ± 0.00	0.00 ± 0.00	0.00 ± 0.00	0.00 ± 0.00	0.00 ± 0.00	0.00 ± 0.00	0.00 ± 0.00	0.00 ± 0.00	27.49 ± 2.91	29.55 ± 1.74
PA	0.00 ± 0.00	0.00 ± 0.00	0.00 ± 0.00	0.00 ± 0.00	0.00 ± 0.00	53.89 ± 1.89	10.34 ± 1.09	0.00 ± 0.00	59.30 ± 3.49	50.25 ± 1.76	0.00 ± 0.00	67.63 ± 3.98	80.92 ± 2.83
HA	34.22 ± 2.71	34.22 ± 2.71	129.95 ± 13.75	73.79 ± 7.81	71.07 ± 4.18	147.15 ± 15.57	79.43 ± 8.41	73.83 ± 7.81	148.63 ± 15.73	75.51 ± 7.99	79.90 ± 8.45	164.83 ± 17.44	79.43 ± 8.40
OA	67.02 ± 2.61	67.02 ± 2.61	153.57 ± 15.58	89.91 ± 5.29	69.21 ± 7.32	149.83 ± 5.24	87.38 ± 8.87	72.02 ± 7.31	181.34 ± 10.66	115.25 ± 4.03	72.77 ± 2.55	196.14 ± 11.53	115.77 ± 4.05
DA	72.24 ± 3.72	72.24 ± 3.72	128.37 ± 7.55	83.04 ± 8.43	79.43 ± 8.41	131.43 ± 4.60	87.39 ± 5.14	88.96 ± 5.23	156.12 ± 15.84	90.21 ± 3.16	89.26 ± 3.12	213.65 ± 21.68	143.82 ± 5.03
MKE													
2H	1.83 ± 0.07	1.83 ± 0.07	33.78 ± 1.99	36.48 ± 3.86	8.00 ± 0.28	51.08 ± 5.18	64.19 ± 3.77	14.35 ± 0.84	60.03 ± 6.35	74.73 ± 7.58	23.99 ± 2.43	100.57 ± 10.64	237.28 ± 24.08
2N	4.83 ± 0.13	4.83 ± 0.13	47.83 ± 5.06	77.43 ± 4.55	15.52 ± 1.57	118.50 ± 12.03	128.95 ± 13.65	64.11 ± 6.78	143.63 ± 8.45	159.54 ± 16.19	102.93 ± 10.45	411.78 ± 24.22	710.11 ± 72.07
MES													
BM	22.02 ± 1.60	22.02 ± 1.60	36.50 ± 1.28	69.62 ± 2.44	24.93 ± 0.87	38.67 ± 4.09	119.48 ± 4.18	21.86 ± 0.76	40.85 ± 0.43	146.28 ± 15.48	28.75 ± 3.04	54.92 ± 1.92	431.29 ± 45.64
HM	73.60 ± 2.99	73.60 ± 2.99	189.60 ± 19.24	262.42 ± 26.63	72.70 ± 7.38	149.83 ± 15.86	274.49 ± 27.86	71.74 ± 7.28	143.60 ± 14.57	341.16 ± 36.10	72.19 ± 7.64	142.64 ± 14.48	1032.01 ± 109.22
OM	46.98 ± 3.33	46.98 ± 3.33	116.56 ± 4.08	128.62 ± 4.50	41.89 ± 2.46	119.84 ± 7.05	307.05 ± 10.75	44.24 ± 1.55	114.66 ± 4.01	473.81 ± 27.86	42.04 ± 2.47	118.80 ± 4.16	723.54 ± 42.55
DM	61.99 ± 6.07	61.99 ± 6.07	77.21 ± 8.17	79.88 ± 5.42	63.04 ± 6.40	135.12 ± 4.73	150.85 ± 15.96	62.84 ± 6.65	136.29 ± 9.24	188.74 ± 6.61	62.67 ± 2.19	139.23 ± 9.44	431.29 ± 15.09
	All groups	CG	II-20	II-40	CG	II-20	II-40	CG	II-20	II-40	CG	II-20	II-40
ACI													
AA	0.00 ± 0.00	0.00 ± 0.00	0.00 ± 0.00	0.00 ± 0.00	0.00 ± 0.00	0.00 ± 0.00	0.00 ± 0.00	0.00 ± 0.00	0.00 ± 0.00	0.00 ± 0.00	0.00 ± 0.00	111,29 ± 6,54	661.95 ± 70.05
PA	0.00 ± 0.00	0.00 ± 0.00	0.00 ± 0.00	0.00 ± 0.00	0.00 ± 0.00	0.00 ± 0.00	0.00 ± 0.00	0.00 ± 0.00	0.00 ± 0.00	50,57 ± 2,97	0.00 ± 0.00	82,28 ± 2,88	537.50 ± 31
HA	34.22 ± 2.71	34.22 ± 2.71	72.00 ± 7.62	44.42 ± 2.61	71.07 ± 4.18	104.15 ± 11.02	72.93 ± 7.72	73.83 ± 7.81	398.93 ± 42.22	76.00 ± 8.04	79.90 ± 8.45	401.18 ± 42.46	486.59 ± 51.
OA	67.02 ± 2.61	67.02 ± 2.61	97.39 ± 3.41	55.20 ± 5.84	69.21 ± 7.32	103.03 ± 6.06	117.79 ± 11.95	72.02 ± 7.31	761.95 ± 77.33	116.30 ± 6.84	79.90 ± 8.45	435.70 ± 15.25	393.75 ± 23
DA	72.24 ± 3.72	72.24 ± 3.72	91.01 ± 3.18	87.60 ± 9.27	79.43 ± 8.41	96.36 ± 9.78	139.00 ± 8.17	88.96 ± 5.23	570.01 ± 33.52	90.00 ± 9.13	89.26 ± 3.12	409.76 ± 14.34	243.48 ± 24.71
MKE													
2H	1.83 ± 0.07	1.83 ± 0.07	28.61 ± 2.90	8.67 ± 0.30	8.00 ± 0.28	36.18 ± 3.83	30.14 ± 1.77	14.35 ± 0.84	166.69 ± 9.80	75.00 ± 7.94	23.99 ± 2.43	245.42 ± 24.91	80.16 ± 8.
2N	4.83 ± 0.13	4.83 ± 0.13	59.87 ± 6.08	17.13 ± 1.74	15.52 ± 1.57	68.91 ± 4.05	78.71 ± 8.33	64.11 ± 6.78	155.05 ± 16.41	161.81 ± 9.52	102.93 ± 10.45	192.72 ± 19.56	103.80 ± 6.
MES													
BM	22.02 ± 1.60	22.02 ± 1.60	46.76 ± 4.95	55.44 ± 1.94	24.93 ± 0.87	77.21 ± 2.70	135.56 ± 4.74	21.86 ± 0.76	78.82 ± 2.76	146.00 ± 5.11	28.75 ± 3.04	86.81 ± 9.19	142.92 ± 5.
HM	73.60 ± 2.99	73.60 ± 2.99	210.21 ± 22.25	223.50 ± 22.68	72.70 ± 7.38	337.90 ± 34.29	283.34 ± 28.76	71.74 ± 7.28	190.89 ± 19.37	341.00 ± 34.61	72.19 ± 7.64	187.25 ± 19.82	243.22 ± 24
OM	46.98 ± 3.33	46.98 ± 3.33	100.33 ± 5.90	109.42 ± 6.43	41.89 ± 2.46	152.41 ± 5.33	127.90 ± 4.48	44.24 ± 1.55	187.81 ± 6.57	469.00 ± 16.41	42.04 ± 2.47	200.41 ± 11.79	255.34 ± 8.
DM	61.99 ± 6.07	61.99 ± 6.07	65.59 ± 2.29	91.32 ± 9.27	63.04 ± 6.40	189.93 ± 12.88	137.44 ± 14.54	62.84 ± 6.65	193.87 ± 20.52	189.57 ± 12.86	62.67 ± 2.19	212.72 ± 7.44	281.17 ± 19

ACI—acids (total), MKE—methyl-ketones (total), MES—methyl-esters (total), AA—acetyl acid, PA—pentanoic acid, HA—hexanoic acid, OA—octanoic acid, DA—decanoic acid, 2H—2-heptanone, 2N—2-nonanone, BM—butanoic methyl ester, HM—hexanoic methyl ester, OM—octanoic methyl ester, DM—decanoic methyl ester.

**Table 4 foods-12-02963-t004:** Eigenvalues of the principal components (correlations) and related statistics.

Values Number	Eigenvalue	% of Variance	Cumulative Eigenvalue	Cumulative Eigenvalue (%)
1	5.499781	49.99801	5.49978	49.9980
2	3.272662	29.75147	8.77244	79.7495
3	1.599775	14.54341	10.37222	94.2929
4	0.307912	2.79920	10.68013	97.0921
5	0.142566	1.29605	10.82270	98.3881
6	0.098043	0.89130	10.92074	99.2794
7	0.044982	0.40893	10.96572	99.6884
8	0.022630	0.20573	10.98835	99.8941
9	0.004761	0.04328	10.99311	99.9374
10	0.004458	0.04052	10.99757	99.9779
11	0.002430	0.02209	11.00000	100.0000

**Table 5 foods-12-02963-t005:** Factorial coordinates of the variables, based on the calculated correlations.

Variable	Factor 1	Factor 2	Factor 3	Factor 4	Factor 5
1. (acetic acid)	−0.448070	0.551151	0.696656	0.033408	−0.038816
2. (pentanoic acid)	−0.525627	0.464894	0.703003	0.078567	0.043553
3. (hexanoic acid)	−0.559014	0.823092	−0.000238	0.005803	0.022618
4. (octanoic acid)	−0.518691	0.733243	−0.385769	−0.150077	−0.052815
5. (decanoic acid)	−0.523602	0.672100	−0.513320	−0.037523	−0.062334
6. (methyl-2-heptanone)	−0.875596	0.070782	−0.391512	0.123622	0.131460
7. (ketony-2-nonanone)	−0.775271	−0.414122	−0.169318	0.424033	−0.001076
8. (methyl-butanoic methyl) ester	−0.832828	−0.509708	0.101182	−0.092405	−0.107786
9. (hexanoic methyl ester)	−0.745704	−0.617916	0.038817	−0.110467	−0.190158
10. (octanoic methyl ester)	−0.787199	−0.493212	0.069796	−0.241167	0.255057
11. (decanoic methyl ester)	−0.964345	−0.156641	0.096700	−0.051665	−0.043134

**Table 6 foods-12-02963-t006:** Discriminant intensity of goat milk taste and smell (mean ± SD).

Features	CG		I-20		I-40		II-20		II-40		SE	*p*-Value
	$\bar{x}$	SD	$\bar{x}$	SD	$\bar{x}$	SD	$\bar{x}$	SD	$\bar{x}$	SD		
Milky taste	3.42	0.58	3.71	0.62	3.75	0.61	3.63	0.65	3.83	0.64	0.06	0.1870
Herbal taste	0.00	0.00	0.21	0.41	0.25	0.44	0.13	0.34	0.17	0.38	0.03	0.1414
Bitter taste	0.08	0.28	0.25	0.53	0.29	0.46	0.21	0.41	0.21	0.41	0.04	0.5335
Sweet taste	1.46	0.51	1.38	0.49	1.33	0.48	1.29	0.46	1.42	0.50	0.04	0.7850
Salty taste	0.75	0.44	0.67	0.48	0.88	0.34	0.83	0.38	0.92	0.28	0.04	0.1852
Sour taste	0.79	0.41	0.75	0.44	0.71	0.46	0.67	0.48	0.67	0.48	0.04	0.8517
Pleasant taste	4.25	0.68	4.42	0.65	4.50	0.66	4.29	0.69	4.63	0.49	0.06	0.2456
Milky smell	3.63	0.65	3.83	0.64	3.92	0.50	3.92	0.65	4.04	0.46	0.05	0.1645
Herbal smell	0.00	0.00	0.17	0.38	0.25	0.44	0.13	0.34	0.21	0.41	0.03	0.1414
Goaty smell	1.21 ^A^	0.41	0.50 ^B^	0.59	0.25 ^B^	0.44	0.46 ^B^	0.66	0.17 ^B^	0.38	0.06	0.0001
Sour smell	0.88	0.34	0.71	0.46	0.58	0.50	0.67	0.48	0.50	0.51	0.04	0.0704
Pleasant smell	4.33 ^A^	0.56	4.38 ^A^	0.58	4.67 ^B^	0.48	4.46 ^B^	0.59	4.75 ^B^	0.44	0.05	0.0281

Abbreviations: CG—control group; I-20—herbal mix I 20 g/day/animal; I-40—herbal mix I 40 g/day/animal; II-20—herbal mix II 20 g/day/animal; II-40—herbal mix II 40 g/day/animal; SD—standard different; SE—standard error; A, B values in rows with different letters are significantly different (*p* < 0.05).

## Data Availability

Data are available upon request from the corresponding author.

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
