# Peer review of "Effect of Herbal Feed Additives on Goat Milk Volatile Flavor Compounds"

_foods, 2023, doi:10.3390/foods12152963_

Round 1

Reviewer 1 Report

To gain further insights into the discrimination of herbal supplements among different treatment groups, it is recommended to complement principal component analysis (PCA) with supervised multivariate analysis techniques, such as partial least squares discriminant analysis (PLS-DA). This approach will allow for a more detailed examination of the contribution of each individual compound towards the differentiation of treatment groups based on volatile data. By incorporating PLS-DA alongside PCA, a comprehensive understanding of the discriminatory power of specific compounds can be achieved, enhancing the interpretation of the results.

Lines 36-38: Remove the sentence and revise the whole abstract.

Line 160: Change "maise" to "maize."

Line 356: Decrease the font size of Table 3 for better viewing and reading.

Line 479: Separate Fig. 10 into individual treatments for better understanding and reading.

Reviewer 2 Report

The authors describe how the inclusion of a herbal supplement in the diet of dairy goats reduces the goaty taste of the milk. Diets, and methods are well described.

However, the results are simply described, and there is little statistical analysis to support part of the conclusion, such as the identification of the volatile components to which the reduction of the goat flavour of the milk is attributed.

The significance of the differences between the different parameters measured, chemical composition of the milk, fatty acids, is neither presented nor discussed.

More effort is needed to discuss the possible reasons leading to the conclusion of the effect of herbal supplements.

It is not necessary to explain how a principal component analysis is carried out (e.g. lines 398 to 406), but to discuss the results further.

The work is interesting, but the results need to be discussed and analysed in more depth.

Reviewer 3 Report

This manuscript details the changes in composition, odor, and taste of milk from goats fed different herbal supplements. The topic is original and closes a gap in our knowledge concerning effects of goat feed. The paper provides information on the subject that has not been obtained previously. It is apparently the first study of its kind. The procedures, results, and interpretations appear to be sound.  Methodology appears to be adequate and the use of controls is appropriate. Conclusions are consistent with the data and the discussion and address the hypothesis. References are appropriate and no additional ones are needed.

Comments:

Line114: Insert x after 800

Lines 114, 116, 135, 217: Use SI units, mo for months, d for days, wk for weeks, s for sec

Line 207: mL

Line 274: 4.36 instead of 4,36

Table 5, Figure 9, and lines 418 and 450: should be methyl-2-heptanone and methyl butanoic

Figure 9 and line 436: should be keto-2-nonanone

Line 436: Delete (?) 

Most of the figures cover data found in Table 3. Figures 1-7 show data presented in Table 3. 
